# An investigation of plasma interleukin-6 in sport-related concussion

**Alex P. Di Battista**[1,2☯]*, **Shawn G. Rhind**[1,2☯], **Doug Richards**[1,3☯], **Michael G. Hutchison**[1,3,4☯]

**1** Faculty of Kinesiology & Physical Education, University of Toronto, Toronto, ON, Canada, **2** Defence Research and Development Canada, Toronto Research Centre, Toronto, ON, Canada, **3** David L. MacIntosh Sport Medicine Clinic, Faculty of Kinesiology & Physical Education, University of Toronto, Toronto, ON, Canada, **4** Keenan Research Centre for Biomedical Science of St. Michael's Hospital, Toronto, ON, Canada

☯ These authors contributed equally to this work.
\* dibattista.alex@gmail.com

**Data Availability Statement:** The corresponding dataset for this paper is available via the Harvard Dataverse (https://doi.org/10.7910/DVN/THZCRP).

## Abstract

### Background

Increasing evidence suggests inflammation is an important component of concussion pathophysiology. However, its etiology, restitution, and potential clinical repercussions remain unknown. The purpose of the current study was to compare the blood concentrations of interleukin (IL) -6, a prominent inflammatory cytokine, between healthy athletes and athletes with a sport-related concussion (SRC), while addressing the potential confounds of sex, recent physical activity, and the interacting effect of concussion history.

### Method

A prospective, observational cohort study was conducted on athletes at a single academic institute participating across 13 interuniversity sports. Follow-up of 96 athletes who agreed to provide a blood sample was completed: 41 athletes with a physician diagnosed SRC, and 55 healthy athletes. Ella™, the high sensitivity immunoassay system by ProteinSimple was used to measure peripheral plasma concentrations of IL-6 within the first week (median = 4 days, range = 2–7) following injury. A resampled ordinary least squares regression was used to evaluate the relationship between IL-6 concentrations and concussion status, while partial least squares regression was used to evaluate the relationship between IL-6 and both symptom burden and time to clinical recovery.

### Results

Regression analysis identified a negative relationship between plasma IL-6 concentrations and the interaction between an acute SRC and a history of concussion (β = -0.29, p = 0.029). IL-6 did not differ between healthy athletes and those with an acute SRC independent of concussion history, and was not correlated with either recovery time or symptom burden in athletes with SRC.

**Funding:** The study was funded by the Canadian Institute of Military and Veteran Health Research, received by MGH (CIMVHR - Task 7).

**Competing interests:** The authors have declared that no competing interests exist.

## Conclusion

Perturbations to circulating IL-6 concentrations, a key inflammatory cytokine, may be more pronounced following SRC in athletes who have a history of concussion. These results add to a growing body of evidence supporting the involvement of inflammation at all phases of recovery following SRC, and potentially support a concomitant effect of prior concussion on acute SRC pathophysiology.

## Introduction

Concussion is a form of mild traumatic brain injury (mTBI) that can elicit numerous symptoms encompassing impairments to cognition, vision, balance, sleep and emotion [1]. While our understanding of the underlying biological mechanisms remains limited, substantial progress has been made in recent years. For example, in 2014 Giza and Hovda's pathophysiological model of concussion was updated to included inflammation as a salient component to secondary injury [2, 3]. While this was based on experimental studies using animal models [3], recent human evidence has shown alterations in inflammatory indices in the acute [4–6], subacute [5–8] and chronic [8–10] phases following injury. As immunological sequelae have the potential to affect mechanisms related to the restitution and/or exacerbation of injury, it is likely that inflammatory signalling following concussion has direct implications on both symptom burden and clinical recovery [11].

In view of the recent progress made in concussion-related inflammation, our group recently identified higher levels of chemokines' monocyte chemoattractant (MCP)-4 and macrophage inflammatory protein (MIP)-1β in the blood of athletes within the first week following a sport-related concussion (SRC), as well as a correlation between recovery time and both MCP-1 & -4 [7]. Changes in peripheral leukocyte mRNA have also been observed at both the acute (six hours) and subacute (seven days) phases following SRC; these changes encompassed both upregulated (i.e., toll-like receptors -1, -2, -4, -7, -8, -10) and downregulated inflammatory genes (i.e., IL-6, chemokine ligand 5, and interferon beta) compared to athlete's own baseline pre-injury gene levels [5]. Moreover, both IL-1RA & -6 have been observed in higher concentrations in the serum of football players acutely following SRC [4]. In addition to these findings in the peripheral blood, neuroinflammation was recently observed in the brains of patients in the subacute (1-2weeks) and chronic (3–4 months) period following an mTBI [8]. Notably, the latter finding of chronic inflammation following either SRC or mTBI has also been supported by blood biomarker research: chemokines have been observed in higher concentrations in athletes with a history of multiple concussions compared to those without a history of concussion [10], and coated platelet levels–an inflammatory surrogate–were positively correlated with the number of prior mTBI's in combat subjects sampled up to years following their last injury [9]. Hence, while there is evidence to suggest that inflammation is involved at all stages of concussion and mTBI recovery, and potentially persists beyond clinical recovery, the mechanisms by which inflammation can mediate recovery, symptom burden, or brain restitution are still unknown.

IL-6 is among the most important and well-studied cytokines in medicine, known historically for its role as a driver of the acute-phase response [12–17]. It can be either pro- or anti-inflammatory depending on the target and receptor binding pathway, and is released from numerous leukocyte subsets during inflammatory activation [13, 14]. As one of the primary mediators of inflammatory signaling, IL-6 is involved in a plethora of medical conditions

including cardiovascular disease [18], diabetes [19], cancer [20], rheumatoid arthritis [21], and mental illness [22–25]. In addition, IL-6 has been found in higher concentrations of the blood [26] and cerebral spinal fluid [27, 28] of patients following severe TBI. Yet, very few studies have evaluated IL-6 concentrations in the blood following SRC. As previously mentioned, downregulated IL-6 mRNA and an increase in circulating cytokine levels have been observed acutely following SRC [4, 5]. While these findings are intriguing, the former looked at mRNA expression in lymphocytes as opposed to circulating protein concentrations, and in the latter, IL-6 was only evaluated in male football players; it is unclear if these findings extend across different sports and in female athletes. This is particularly relevant given the noted differences in immune function [29, 30] as well as both concussion symptom burden and recovery trajectories between males and females [1, 31, 32]. Furthermore, as concussion history may be associated with perturbed inflammatory signaling [9, 10], its influence on IL-6 concentrations following acute injury are unclear. Hence, given these knowledge gaps, and the importance of IL-6 in many facets of immune signaling, a fulsome understanding of its role in inflammation following concussion is required.

The purpose of this study was to examine the blood concentrations of IL-6 in response to sport-related concussion while addressing the potential confounds of sex, recent physical activity, and the interacting effect of concussion history. In addition, we sought to evaluate the relationship between IL-6 and clinical indices (days to medical clearance and symptom burden) within the SRC group. We hypothesized that IL-6 concentrations would be higher in the blood of athletes with SRC compared to healthy athletes irrespective of concussion history and would correlate with higher symptom burden and longer recovery times.

## Methods

### Participants

A sample of 96 athletes (n = 43, male; n = 54, female) from 13 interuniversity sports at a single academic institution were recruited for this study: basketball (n = 11), field hockey (n = 6), figure skating (n = 1), football (n = 3), ice hockey (n = 26), lacrosse (n = 9), mountain biking (n = 2), rowing (n = 1), rugby (n = 18), soccer (n = 4), softball (n = 1), swimming (n = 1), and volleyball (n = 13). Fifty-five healthy athletes (n = 21, male n = 34, female) were enrolled prior to the beginning of their competitive season. Forty-one athletes were enrolled within seven days (median = 4, range = 2–7) of a sport-related concussion (SRC). Concussion diagnosis and medical clearance decisions were provided by a staff physician at the university sport medicine clinic in accordance with the most recent Concussion in Sport Group guidelines [1]. Briefly, medical clearance for unrestricted activity (clinical recovery) required athletes to be asymptomatic at rest and to have completed a graded return-to-play (RTP) protocol. The RTP protocol consisted of light aerobic exercise, more intensive training, sports-specific exercises, noncontact practice participation, and finally, high-risk practice. Following this, the physician verified that the athlete had returned to their baseline level of cognitive and neurological functioning. Athletes were excluded if they were suffering from a non-head sports injury, or if their most recent prior concussion occurred within six months of study participation. Prior to enrollment, all participants provided written informed consent; all study procedures were in accordance with the declaration of Helsinki, and approved by the Health Science Research Ethics Board, University of Toronto (protocol reference # 27958).

### Blood acquisition and plasma IL-6 measurement

At the time of enrollment, participant blood was drawn by standard venipuncture into a 10-mL K$_2$EDTA tube, equilibrated at room temperature for one hour, and centrifuged for two

minutes using a PlasmaPrep 12$^{\text{TM}}$ centrifuge (Separation Technology Inc., FL, USA). After centrifugation, plasma was aliquoted and stored at -80˚C until analysis. Blood draws were not performed on subjects who were knowingly symptomatic with a viral or bacterial infection, seasonal allergies, or on any medication other than birth control at the time of venipuncture. Interleukin-6 was measured using the Ella™ system (ProteinSimple, Biotechne, San Jose, CA, USA). The assay was run in triplicate, according to the manufacturer's directions with no deviations, and the resultant blood analyte concentrations were derived through calculating the mean of the triplicate values.

## Symptoms

Participant symptoms were determined via a 27-item post-concussion symptom scale using a seven-point Likert rating as part of the C3 Logix concussion assessment tool. C3 Logix symptoms are based on the 22-item scale as part of the Sport Concussion Assessment Tool (SCAT)-5 [1], with the addition of five questions: "ringing in the ears", "numbness or tingling", "sleeping more than usual", "sleeping less than usual", and "difficulty sleeping soundly". The SCAT is the most widely used tool to assist in the diagnosis, management, and prognosis of individuals with concussion [1], and has displayed reliability and validity for assessing symptom presence and severity [31, 33]. A total symptom score was obtained by summing the presence or absence of each symptom irrespective of severity, with a maximum value of 27; symptom severity was evaluated by summing the rated symptom score for each symptom. In addition, four distinct symptom clusters were created through summing the symptom scores related to somatic complaints (headache, pressure in head, neck pain, nausea/vomiting, dizziness, blurred vision, balance problems, sensitivity to light, sensitivity to noise, ringing in the ears, numbness or tingling), cognition (feeling slowed down, feeling in a fog, don't feel right, difficulty concentrating, difficulty remembering, confusion), sleep (fatigue/low energy, drowsiness, trouble falling asleep, sleeping more than usual, sleeping less than usual, difficulty sleeping soundly), and emotion (more emotional, irritability, sadness, nervous/anxious). This approach has been previously employed by our group with the 22-item SCAT symptom questionnaire [7, 34], and has been updated to include the additional five symptom scores included from C3 Logix.

## Statistics

IL-6 values were statistically analyzed only if they fell above the manufacturer provided lower limit of detection (0.26 pg/mL) and had a coefficient of variation (CV) between replicates of < 20%. Of the 96 participants included in the study, all IL-6 participant samples met these criteria; one IL-6 value of 35.9 pg/mL was deemed an outlier (> 10 standard deviations (SD) above the mean) and was thus replaced with an imputed median value. The final range of IL-6 values was 0.53–6.55 pg/mL, with an average CV between replicates of 5.9% (range = 0.4–18.1).

Variables missing no more than 20% of their total values were imputed using the variable median. After imputation and prior to statistical analysis, all variables were tested for deviations from normality by calculating sample skewness and kurtosis, with percentile p-values obtained by comparison against a simulated null distribution (random gaussian noise, 1000 simulated samples). Kurtosis ranged from 1.2 (p = 1.00) to 8.6 (p < 0.001), and skewness ranged from -0.7 (p = 0.002) to 2.2 (p < 0.001). Variables that violated normality were transformed via two-tailed winsorization (10%).

Characteristic variables found in **Table 1** were univariately compared between healthy athletes (n = 55) and athletes with SRC (n = 41) by calculating the mean differences between

groups for each variable, followed by bootstrapping the mean difference scores (1000 resamples) to obtain standardized effect size in terms of bootstrap ratios (BSR; mean / standard error) and empirical p values based on the bootstrap estimates of the standard error, which were corrected at a false discovery rate (FDR) of 0.05. Categorical characteristic variables were evaluated by $\chi2$.

The primary aim of this study was to evaluate the relationships between IL-6 and concussion, while controlling for potential confounding variables (time since recent physical activity, sex), and evaluating the potential influence of a prior history of concussion on IL-6 concentrations in acutely concussed individuals. To do this, an ordinary least squares (OLS) multilinear regression was employed, with participant IL-6 values as the outcome variable, and sex, time since recent physical activity, acute concussion, and an interacting variable (acute concussion * history of concussion), as the predictor variables; all data was mean-centred prior to analysis. The model was run in a bootstrap resampling framework (5000 iterations). Mean coefficient values were calculated on the sampling distribution, from which a BSR was derived. Percentile p values were identified for each variable by computing the fraction of bootstrapped coefficient values not enclosing zero effect in a two-tailed framework. An Adjusted mean model $R^2$ value and 95% confidence intervals (CI) were also calculated.

**Table 1. Athlete characteristics.**

| Variable | SRC (n = 41) | Healthy (n = 55) |
|---|---|---|
| Age (years) | 20.9 (19.6–22.0) | 21 (19–22) |
| Sex (n, % male) | 21 (51.2) | 21 (38.2) |
| Sport (n, %) | | |
| Basketball | 1 (2.4) | 10 (18.2) |
| Field Hockey | 1 (2.4) | 5 (9.1) |
| Figure Skating | 1 (2.4) | -- |
| Football | 3 (7.3) | -- |
| Ice Hockey | 9 (21.9) | 17 (30.9) |
| Lacrosse | 3 (7.3) | 6 (10.9) |
| Mountain Biking | 2 (4.9) | -- |
| Rowing | 1 (2.4) | -- |
| Rugby | 15 (36.6) | 3 (5.4) |
| Soccer | 1 (2.4) | 3 (5.4) |
| Softball | 1 (2.4) | -- |
| Swimming | 1 (2.4) | -- |
| Volleyball | 2 (4.9) | 11 (20) |
| C3 Logix | | |
| Total Symptoms | 13.5 (10–18) | 2 (0–4) |
| Symptom Severity | 31 (14.2–43.8) | 3 (0–7) |
| Days to blood sample from injury | 4 (3–5) | -- |
| Days to medical clearance | 27 (22–51) | -- |
| Concussion History (n, %) | 20 (48.8) | 21 (38.2) |
| Time since last concussion (years) | 3.1 (1.5–5.4) | 4.1 (2.3–5.9) |
| IL-6 (pg/mL) | 1.3 (1.1–1.8) | 1.3 (1.1–2) |

SRC, sport related concussion; SCAT, sport concussion assessment tool

All values reported as the median and interquartile range, unless otherwise stated

*SCAT5 scores at the time of blood draw

A correlational partial least square regression (PLSC) was used to evaluate the relationship between IL-6 values and clinical indices of concussion (symptom burden, time to medical clearance) in athletes with SRC. PLSC identifies the maximal covariance between an outcome variable (IL-6) and a set of predictor variables (symptoms, time to medical clearance), and is well posed to handle multicollinearity between predictor variables through the creation of orthogonal, latent variables [35, 36]. The PLS was used in a bootstrap resampling framework (5000 iterations) to generate sets of weighted mean variable loadings and corresponding BSRs, with percentile p values identified for each loading, corrected at a false discovery rate of p = 0.05. An out-of-sample, leave-two-out cross correlation ($R^2$) value was also calculated on the PLS model.

All statistical analyses were performed with R (RStudio, version 1.2.1335, Boston, United States).

## Results

Athlete demographic and characteristic variables can be seen in **Table 1**. Athletes with SRC reported significantly higher scores compared to healthy athletes on all symptom variables measured: total symptoms (BSR = 8.3, p < 0.001) calculated symptom severity (BSR = 7.5 p < 0.001) and all symptom clusters (cognitive, BSR = 7.1, p < 0.001; somatic, BSR = 9.2, p < 0.001; sleep, BSR = 4.8, p < 0.001; emotion, BSR = 2.6, p = 0.01). However, there were no differences in plasma IL-6 concentrations (healthy = 1.3 pg/mL; SRC = 1.3 pg/mL, BSR = 0. 6, p = 0.569).

OLS regression results evaluating the relationship between IL-6 and numerous predictors across all subjects can be seen in **Table 2**. The presence of an acute concussion alone was not significantly correlated with IL-6 concentrations. However, the OLS model identified a significant negative relationship between IL-6 concentrations and the interaction of an acute concussion with a history of a prior concussion(s), while controlling for the potential effects of sex and recency of physical activity (β-Coefficient = -0.29, BSR = 2.4, p = 0.019).

PLS regression analysis evaluating the correlation between IL-6 concentrations and clinical indices (symptom burden and time to medical clearance) in athletes with SRC can be seen in **Table 3**. Twelve athletes were lost to follow-up, and thus their medical clearance data was not available. Hence, the PLS analysis was run on 29 athletes with SRC. There was no significant correlation between IL-6 and either symptom burden or days to medical clearance. Concussion history was significantly negatively correlated with IL-6 (BSR = 3.5, p = 0.004).

## Discussion

The primary finding of this study was that perturbations to circulating IL-6 were more pronounced following SRC in athletes who have a history of concussion. Yet, IL-6 did not differ between healthy athletes and those with an acute SRC independent of concussion history, and we did not observe a correlation between IL-6 and either recovery time or symptom burden in injured athletes.

We found that the combination of having both an acute SRC and a history of concussion was associated with lower IL-6 concentrations in athletes, while there was no correlation between IL-6 concentrations and symptom reporting or time to clinical recovery. This is contrary to prior findings by Nitta and colleagues of an acute increase in serum IL-6 that was correlated with symptom duration [4]. However, differing sample acquisition times may help describe the apparent disparity, as the aforementioned study reported an increase within 6h of injury. In contrast, in the current study our first sample was taken at a median of four days following injury (range = 2–7; correlation between IL-6 and days from injury to blood draw: $R^2$ =

**Table 2. OLS regression between IL-6 plasma levels and multiple predictors.**

| Variables | β-Coefficient | SE | 95% CI | BSR | Percentile P Value |
|---|---|---|---|---|---|
| Sex | 0.04 | 0.16 | -0.17–0.23 | 0.3 | 0.793 |
| Recent PA | -0.03 | 0.08 | -0.27–0.016 | 0.5 | 0.674 |
| Acute Conc. | 0.33 | 0.21 | -0.04–0.48 | 1.6 | 0.101 |
| Acute Conc. * Hx of Conc. | -0.29 | 0.13 | -0.48 – -0.05 | 2.4 | **0.019** |

**Adjusted model $R^2$ value** = 0.05; **95%CI** = -0.02–0.16

OLS, ordinary least squares; IL-6, interleukin-6; SE, standard error; CI, confidence interval; BSR, bootstrap ratio; Conc, Concussion; Hx, History.

Acute concussion was defined as having been injured within seven days (median 4) of blood sampling.

Recent physical activity was defined by rank numerical order of having exercised within a given timeframe: zero ($\leq$ 4 hrs), one (4 $\geq$12 hrs), two (12 > 24hrs), and three ($\geq$ 24hrs). Concussion history is coded as zero (no prior concussion history), one (one prior concussion), and two (multiple prior concussions).

All data were mean-centred prior to analysis.

The mean β-coefficient values, 95% CI and BSRs were derived by bootstrapped resampling (5000 iterations).

P values were derived by calculating the fraction of bootstrapped coefficient values not enclosing zero effect in a two-tailed framework.

0.09, p = 0.53). However, our null findings at 2–7 days following injury are in agreement with these authors, who also observed no differences in IL-6 concentrations between healthy athletes and those with SRC by 48h post-injury, and is also supported by prior findings of lower IL-6 mRNA in athletes within 6h following SRC that returned to baseline levels by day seven [5]. Furthermore, the aforementioned study by Nitta and colleagues measured a cohort of male football players, while the current study helps extend these observations into sports beyond football and across both males and females. Most importantly, perturbations to IL-6 in the present study required both the presence of an acute concussion and a history of prior concussions, and it is unclear how the results of the aforementioned studies may have been augmented by evaluating the interaction between acute SRC and a history of concussion.

**Table 3. PLS regression between IL-6 and clinical indices in athletes with SRC.**

| Variables | Loadings | SE | 95% CI | BSR | P Value | FDR |
|---|---|---|---|---|---|---|
| Sex | 0.31 | 0.23 | -0.23–0.67 | 1.3 | 0.207 | No |
| Concussion History | -0.58 | 0.16 | -0.86 – -0.23 | 3.5 | **0.004** | **Yes** |
| Days to Medical Clearance | 0.15 | 0.21 | -0.23–0.59 | 0.7 | 0.508 | No |
| Total Symptoms | 0 | 0.21 | -0.36–0.33 | 0 | 0.988 | No |
| Symptom Severity | 0.03 | 0.24 | -0.37–0.39 | 0.1 | 0.916 | No |
| Cognitive | -0.18 | 0.25 | -0.54–0.30 | 0.7 | 0.564 | No |
| Somatic | 0.17 | 0.22 | -0.26–0.50 | 0.8 | 0.527 | No |
| Emotion | 0 | 0.28 | -0.50–0.45 | 0 | 0.983 | No |
| Sleep | 0.12 | 0.22 | -0.34–0.47 | 0.5 | 0.627 | No |

**Adjusted model $R^2$ value** = 0.07; **95%CI** = 0.04–0.11

PLS, partial least squares; IL-6, interleukin-6; SE, standard error; CI, confidence interval; BSR, bootstrap ratio; FDR, false discovery rate.

Acute concussion was defined as having been injured within seven days (median 4) of blood sampling.

Concussion history is coded as zero (no prior concussion history), one (one prior concussion), and two (multiple prior concussions).

All data were mean-centred and scaled (z-score transform) prior to analysis.

The mean weighted variable loading values, 95% CI and BSRs were derived by bootstrapped resampling (5000 iterations).

P values were derived by calculating the fraction of bootstrapped loading values not enclosing zero effect in a two-tailed framework, corrected at a false discovery rate of p = 0.05.

Model $R^2$ derived via a leave-two-out cross correlation on 1000 resamples.

That we did not see a difference in IL-6 concentrations between healthy athletes and those with an SRC independent of concussion history is not in agreement with our hypothesis, and is seemingly opposed to our previous findings of higher concentrations of inflammatory markers in the blood of athletes following SRC [7]. However, our hypothesis was guided by limited research in this space to date, and the salient inflammatory correlates identified by our prior work were chemoattractant molecules (MCP-1,-4, and MIP-1β) as opposed to traditional inflammatory cytokines [7, 10]. Furthermore, our prior work did not evaluate the interaction between a history of concussion and an acute concussion. Indeed, the phenomenon known as "immunoexcitotoxicity", posits that prior trauma to the brain can prime resident microglial cells–immune cells present in the CNS–allowing for a potentially exaggerated inflammatory response when faced with subsequent trauma [37]. While this theory may extend into peripheral immune signaling and thus explain the requirement of both an acute SRC and a history of concussion to observe a difference in the inflammatory (IL-6) response to concussion, it does not provide an obvious answer as to why IL-6 concentrations were lower. While we can assume that not having samples within 24h of injury may have resulted in a missed initial proinflammatory state, lower IL-6 concentrations may indirectly suggest pro-inflammation as it can act as an anti-inflammatory agent in certain situations [12, 14, 16]. Indeed, attempts at interpretation are complicated both by this duality and previous studies which have found a correlation between IL-6 and improved recovery in severe TBI [38, 39]. In sum, any interpretations are speculative, and the underlying mechanisms tying IL-6 to concussion pathophysiology require future investigation. Yet, our results add to a growing body of literature that suggest biological disturbances may extend beyond the acute and subacute phases following concussion, potentially for months-to-years; history of concussion in the current study was calculated based on athletes reporting of their last concussion, which was on average 3–4 years from the time of study participation.

It is plausible that having a history of concussion as an athlete may carry a burden of psychological stress, which can augment immune function [24, 25, 40]. This may in-part be due to the abundance of media attention given to the potential link between repetitive concussion history and future neurodegeneration in sport [41, 42]. As we did not evaluate metrics of athlete's subjective interpretation on the impact of their past and present injuries to their psychological well-being or pre-existing clinical anxiety disorders, we cannot speculate on this. However, we feel that the interplay between anxiety, concussion and the biological underpinnings of recovery is a fruitful avenue for future research.

In the present study, we were able to obtain 100% detectability using the manufacturer's suggested lower limit of detection (LLOD) of 0.26 pg/mL on the Ella™ platform (ProteinSimple, USA). This is promising, as previous attempts to measure plasma IL-6 by our group were unable to achieve concentrations above the LLOD (median of 0.06 pg/mL) in greater than 50% of the samples analyzed using the mesoscale discovery™ platform [6, 10, 44]. Our prior results are not surprising given that the MSD (MSD VPlex™, proinflammatory panel 1) product insert suggests 37% detectability in the plasma of healthy humans [45]. However, the aforementioned study that evaluated IL-6 in the serum of athletes following SRC observed 88.3% detectability in their sample of participants using a similar assay on the MSD platform [4]. These differences in analyte detectability between studies may represent differences in acquisition time-points and/or the general variability between instruments and assays both within and across platforms. Yet, this serves to reiterate the importance of larger, multisite, multiplatform studies to aid in reproducibility and validation in the field of biomarker research, particularly with injures such as SRC where there is substantial heterogeneity in pathophysiology, relatively small shifts in biology, and an abundance of potential confounds.

The current study must be interpreted within the context of its limitations. While we controlled for sex, our sample size was underpowered to separately evaluate IL-6 in males and females following SRC, including their potential relationships to symptom burden and recovery time. Despite not finding differences in IL-6 concentrations between males and females in the current study, there are noted sex differences in immune function generally [29, 30], and concussion indices specifically [31, 32]. In particular, we recently evaluated a suite of inflammatory markers (not including IL-6) in male and female athletes following SRC and found profound differences in their relationship to symptom presentation [43]. Furthermore, as previously mentioned, given prior observed relationships between IL-6, psychological stress and mental health [22, 23, 25, 44–47], subset analyses evaluating the contribution of these variables to our findings would ostensibly help in clarifying the degree to which IL-6 differs in subjects as a consequence of concussion as opposed to a potential tangential stressor which may or may not be consequent to their injury. Finally, elucidation of the underlying mechanism(s) implicating IL-6 signalling to concussion would benefit from a combinatorial approach utilizing advanced neuroimaging alongside an evaluation of multiple inflammatory genes and proteins. This approach would provide insight into the complexity of inflammatory signaling while appreciating the context of these biological signatures in relation to neuroanatomical perturbations. Yet, despite these limitations, the results of the current study were able to identify a robust correlation between IL-6 and the interaction between an acute concussion and a history of concussion, that was not influenced by sex or recent physical activity.

## Conclusion

Perturbations to circulating concentrations of IL-6, a key inflammatory cytokine, are more pronounced following SRC in athletes who have a history of concussion. However, IL-6 does not differ between healthy athletes and athletes sampled within one week of an SRC and does not correlate with symptom burden or time to clinical recovery. These results add to a growing body of evidence supporting the involvement of inflammation at all phases of recovery following SRC, and potentially support a concomitant effect of prior concussion on acute SRC pathophysiology.

## Author Contributions

**Conceptualization:** Alex P. Di Battista, Shawn G. Rhind, Doug Richards, Michael G. Hutchison.

**Data curation:** Alex P. Di Battista, Shawn G. Rhind, Michael G. Hutchison.

**Formal analysis:** Alex P. Di Battista.

**Funding acquisition:** Shawn G. Rhind, Michael G. Hutchison.

**Investigation:** Alex P. Di Battista, Doug Richards, Michael G. Hutchison.

**Methodology:** Alex P. Di Battista.

**Project administration:** Shawn G. Rhind, Doug Richards, Michael G. Hutchison.

**Resources:** Shawn G. Rhind, Doug Richards.

**Supervision:** Doug Richards, Michael G. Hutchison.

**Validation:** Alex P. Di Battista, Michael G. Hutchison.

**Visualization:** Alex P. Di Battista.

**Writing – original draft:** Alex P. Di Battista, Michael G. Hutchison.

**Writing – review & editing:** Alex P. Di Battista, Shawn G. Rhind, Doug Richards, Michael G. Hutchison.

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
