## [Decision Letter · Decision Letter 0]

16 Mar 2020

PONE-D-19-34278

An Investigation of Plasma Interleukin-6 in sport-related concussion.

PLOS ONE

Dear Dr. Di Battista,

Thank you for submitting your manuscript to PLOS ONE. After careful consideration, we feel that it has merit but does not fully meet PLOS ONE’s publication criteria as it currently stands. Therefore, we invite you to submit a revised version of the manuscript that addresses the points raised during the review process.

We would appreciate receiving your revised manuscript by Apr 30 2020 11:59PM. To enhance the reproducibility of your results, we recommend that if applicable you deposit your laboratory protocols in protocols.io, where a protocol can be assigned its own identifier (DOI) such that it can be cited independently in the future. For instructions see: http://journals.plos.org/plosone/s/submission-guidelines#loc-laboratory-protocols

We look forward to receiving your revised manuscript.

Kind regards,

Calogero Caruso, MD

Academic Editor

PLOS ONE

Journal Requirements:

Reviewers' comments:

Reviewer's Responses to Questions

**Comments to the Author**

1. Is the manuscript technically sound, and do the data support the conclusions?

Reviewer #1: Yes

2. Has the statistical analysis been performed appropriately and rigorously? 

Reviewer #1: I Don't Know

3. Have the authors made all data underlying the findings in their manuscript fully available?

Reviewer #1: Yes

4. Is the manuscript presented in an intelligible fashion and written in standard English?

Reviewer #1: Yes

5. Review Comments to the Author

Reviewer #1: This is an interesting study and the motivation of this work is clearly explained. The paper is generally well written. However, in my opinion, the paper is redundant, in particular the results of two papers [ref. 4,5] are repeated several times in the introduction (page 4, line 20-23; page 5, lines 21-23; page 6 lines 1-2) and discussion paragraphs. I therefore recommend restructuring them, making a mention of the previous citations (“as previously mentioned”) and to cite them fewer times in order to avoid redundancy.

Below I have provided some minor remarks on the text.

Please, control all the abbreviations. For example: page 2, line 6, IL-6 abbreviation should be identified at the first appearance in the text; pag 4, line 19, the abbreviation of SRC has already been introduced in pag 2, line 7; pag 5, lines 2-3, the abbreviation of mTBI has already been introduced in pag 4, line 3, pag 5, line 20, spell-out CSF.

Pag 4, lines 20-23. It may be relevant to specify which genes are up-regulated and which down-regulated compared with control (specify which control).

Pag 6, lines 2-4. Please clarify sentence. What do the authors mean by "as opposed to circulating protein concentrations "?

Pag 7, lines 1-5. The total number of participants for individual sport is 97, not 96.

Pag 8, line 2. “Blood” instead “plasma”.

Pag 12, Table 1. Again, the total number of SRC for individual sport is 42, not 41. Please, recalculate the percentages or clarify why a recruited subject has been removed from the total. If necessary, also correct the total number of participants and/or SRC individuals in the abstract.

Please, add a point at the end of the Table 1 title and uniform the descriptive legend with the other two tables, using first the abbreviation and then the spelled-out form.

Pag 15, line 18. Add dash in IL6.

6. PLOS authors have the option to publish the peer review history of their article (what does this mean?). If published, this will include your full peer review and any attached files.

Reviewer #1: No

---

## [Author Response · Author response to Decision Letter 0]

28 Mar 2020

Please see attachment "Response to Reviewers"

---

## [Decision Letter · Decision Letter 1]

7 Apr 2020

An Investigation of Plasma Interleukin-6 in sport-related concussion.

PONE-D-19-34278R1

Dear Dr. Di Battista,

We are pleased to inform you that your manuscript has been judged scientifically suitable for publication and will be formally accepted for publication once it complies with all outstanding technical requirements.

With kind regards,

Calogero Caruso, MD

Academic Editor

PLOS ONE

Additional Editor Comments (optional):

Reviewers' comments:

Reviewer's Responses to Questions

**Comments to the Author**

1. If the authors have adequately addressed your comments raised in a previous round of review and you feel that this manuscript is now acceptable for publication, you may indicate that here to bypass the “Comments to the Author” section, enter your conflict of interest statement in the “Confidential to Editor” section, and submit your "Accept" recommendation.

Reviewer #1: All comments have been addressed

2. Is the manuscript technically sound, and do the data support the conclusions?

Reviewer #1: (No Response)

3. Has the statistical analysis been performed appropriately and rigorously? 

Reviewer #1: (No Response)

4. Have the authors made all data underlying the findings in their manuscript fully available?

Reviewer #1: (No Response)

5. Is the manuscript presented in an intelligible fashion and written in standard English?

Reviewer #1: (No Response)

6. Review Comments to the Author

Reviewer #1: (No Response)

7. PLOS authors have the option to publish the peer review history of their article (what does this mean?). If published, this will include your full peer review and any attached files.

Reviewer #1: No

---

## [Editor Report · Acceptance letter]

14 Apr 2020

PONE-D-19-34278R1 

An Investigation of Plasma Interleukin-6 in sport-related concussion. 

Dear Dr. Di Battista:

I am pleased to inform you that your manuscript has been deemed suitable for publication in PLOS ONE. Congratulations! Your manuscript is now with our production department. 

With kind regards,

on behalf of

Prof. Calogero Caruso 

Academic Editor

PLOS ONE